# FUNCTIONAL FORM FOR SEGMENTATION ACCURACY PREDICTION FOR COMPARING ARCHITECTURES AND ESTIMATING COST-OPTIMAL TRAINING DATASET

## ABSTRACT

Medical image segmentation plays a significant role in pre-treatment diagnosis, treatment, and post-treatment assessment of various medical conditions. To improve image segmentation accuracy, novel architectures are proposed, and their performance is compared with existing architectures to demonstrate superiority; benchmark datasets are used for these comparisons. For conciseness, only accuracies using the full benchmark datasets are reported, although results for subsets of varying sizes could also be readily obtained. However, there is currently no established method for predicting the expected accuracy of machine learning (ML) models as the training dataset size increases. In this work, we propose a procedure for developing functional forms that predict performance as a function of training dataset size and model architecture (with a fixed structure). We empirically demonstrate that these models can estimate performance for any dataset size, compute asymptotic performance, and compare the relative effectiveness of different machine learning (ML) models. Additionally, they can be used to determine the cost-optimal training dataset size. The scope of this study is limited to medical image segmentation models.

## 1 INTRODUCTION

Medical image segmentation models isolate areas of interest within medical images and play a significant role in pre-treatment diagnosis, treatment, and post-treatment assessment of various conditions. Machine learning (ML) models for medical image segmentation are reducing healthcare costs and increasing the efficiency of healthcare providers. However, current ML models remain imperfect, and research for developing better architectures continues.

Major factors—dataset size, dataset quality, and model architecture—determine the performance of a trained ML model. While larger datasets generally improve accuracy, data collection is costly and accuracy gain is sub-linearly with size, making unlimited expansion impractical. Consequently, researchers have largely concentrated on architectural innovation.

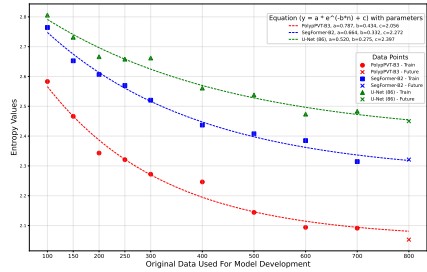

Figure 1: Functional forms for U-Net, SegFormer, and PolypPVT. The entropy of 9 of the 10 trained ML models was used to fit the functional form $y = ae^{-bn} + c$

Since 2015, deep neural network (DNN)-based models, from fully convolutional networks such as U-Net (Ronneberger et al., 2015) to recent transformer- and convolutional attention module (CAM)-based approaches (Rahman & Marculescu, 2023), have steadily advanced segmentation performance. For example, the IoU scores of architectures including U-Net (Ronneberger et al., 2015), UNet++ (Zhou et al., 1807), PraNet (Fan et al., 2020), UACANet-L (Kim et al., 2021), SSFormerPVT (Wang et al., 2022), PolypPVT (Dong et al., 2021), and PVT-CASCADE (Rahman & Marculescu, 2023) have improved from 74.6 to 87.76 on the Polyp segmentation dataset (Jha et al., 2020). This trend of discovering superior architectures is expected to continue.

Typically, ML models are trained, validated, and tested with benchmark datasets: training examples estimate parameters, validation data guide hyperparameter and model selection, and test examples measure final performance (Raschka, 2020; Zheng et al., 2019). To establish superiority, new models are empirically compared with prior work. Yet such evaluations cannot predict performance gains if more training examples were available. If one could estimate improvement as a function of dataset size—and combine it with data collection costs—then cost-optimal dataset sizes could be determined. A functional form developed from empirical observations would enable estimation of expected performance for arbitrary dataset sizes, determination of cost-optimal training sets, and principled comparison of architectures with similar complexity or different parameter counts (see Fig.1, 4).

In this work, we propose a method for developing such functional forms for medical image segmentation models. Our approach converts performance metrics, such as Intersection over Union (IoU), into a probability distribution, which is then used to estimate the entropy of an ML model. Since the transformation from scores to entropy is approximate and potentially non-unique, the resulting functional forms are only approximations. However, as empirical evaluation itself is an approximation of true model performance, we believe our method does not introduce additional uncertainty but instead provides insights unavailable through traditional evaluation.

**Our Contributions:** Our main contributions are summarized as follows: (1) We introduce a novel methodology that converts a trained image-segmentation model's IoU scores into entropy; (2) we develop a functional-form based framework for modeling entropy as a function of training dataset size, where the parameters reveal both performance gains from additional data and asymptotic performance limits; (3) we demonstrate how the selected functional form can be used to compare different architectures (U-Net, SegFormer, PolypPVT) and variants with varying parameter counts, showing strong agreement with observed performance on the Kvasir dataset; (4) through this analysis, we identify exponential decay with an added constant as the most consistent and theoretically justified functional form; (5) we derive an analytical expression to determine cost-optimal dataset size, balancing annotation effort with expected performance gains; (6) finally, we present a heuristic approach to revert entropy values into IoU estimates, thereby extending the interpretability and practical utility of the method.

## 2 PRELIMINARIES AND RELATED WORK

### 2.1 NOTATIONS AND PRELIMINARIES

Let $[n]$ be the set $1, 2, \ldots, n$. Let $p = p_i \mid i \in [n]$ be a discrete probability distribution. Consider dividing the real numbers between 0 and 1 into $k$ intervals, where for $i \in [k-1]$, the range $r_i = [a, b)$, with $a = \frac{i-1}{k}$ and $b = \frac{i}{k}$, and the final range $r_k = \left[\frac{k-1}{k}, 1\right]$. These ranges are used to obtain the probability values $p_i$ from IoU scores. Given a dataset $\mathcal{D} = (x_i, y_i) \mid i \in [N]$, where $x_i$ and $y_i$ are identical-size images: $x_i$ is the original medical image, and $y_i$ is its corresponding binary mask, with pixels of interest labeled as 1 and all other pixels labeled as 0.

For training and evaluation purposes, the $N$ examples in $\mathcal{D}$ are partitioned into two subsets: $\mathcal{D}_t$, which contains $N_t$ examples, and $\mathcal{D}_e$, which contains $N_e$ examples. Clearly, $\mathcal{D}_t \cap \mathcal{D}_e = \emptyset$, and $\mathcal{D}_t \cup \mathcal{D}_e = \mathcal{D}$. In an ideal partition, the distribution of examples in both the training and evaluation datasets is expected to be identical. Typically, $\mathcal{D}_e$ contains between 10% and 20% of the examples. For our study, we use 20% of the examples for the evaluation set and the remaining 80% for the training set. To distinguish between the functional forms developed here and the machine learning models trained with training examples, we refer to the latter as ML models. From $\mathcal{D}_t$, we select smaller subsets to train several ML models. Let $\mathcal{D}^{(n)} \subseteq \mathcal{D}_t$ be a training dataset containing $n$ examples, i.e., $|\mathcal{D}^{(n)}| = n$. A machine learning model developed with training data from $\mathcal{D}^{(n)}$ is referred to as $\text{ML}^{(n)}$.

**Definition 1.** (**IoU score**) Given two sets $\mathcal{X}$ and $\mathcal{Y}$, the IoU score is $IoU = \frac{|\mathcal{X} \cap \mathcal{Y}|}{|\mathcal{X} \cup \mathcal{Y}|}$, where $|\mathcal{X} \cap \mathcal{Y}|$ and $|\mathcal{X} \cup \mathcal{Y}|$ denote the cardinalities of the intersection and union of the sets $\mathcal{X}$ and $\mathcal{Y}$.

**Definition 2.** (**Entropy**) Let $p = \{p_i \mid i \in [n]\}$ be probabilities of events. The entropy is defined as $H(p) = -\sum_{i=1}^{n} p_i \log p_i$, where the logarithm is typically base 2 in information theory, but we use base $e$ in this work.

## 2.2 RELATED WORK

To the best of our knowledge, no prior work directly relates medical image segmentation performance of a given DNN model and a training dataset size. We therefore review studies on classification models and large language models (LLMs), which provide useful directions for our approach.

Neural network performance is known to improve with training time, dataset size, and model size, though techniques such as learning rate schedules, normalization, and regularization can accelerate convergence (Sarkar, 1995; Shen et al., 2024). Prior work on scaling falls into three categories: bias–variance analysis, power-law approximations, and scaling-law-based modeling. Classical bias–variance theory shows error decreases as $O(1/\sqrt{n})$ with sample size (Geman et al., 1992). Empirical studies found error decays exponentially with more data (Cho et al., 2015) and performance often follows logarithmic or power-law trends (Sun et al., 2017; Hestness et al., 2017).

Other work compared functional forms for extrapolating accuracy from small datasets. The biased power law provided the best fit, consistent with the need to account for Bayes error (Johnson et al., 2018; Theisen et al., 2021). Scaling laws have also been extended to model architecture: EfficientNet formalized model size scaling (Tan & Le, 2019), while Rosenfeld et al. (2019) proposed a joint functional form combining model and dataset size.

These insights are valuable but not directly transferable to medical image segmentation, motivating the need for specialized modeling in this domain.

## 3 DEVELOPMENT OF FUNCTIONAL FORM

Entropy has been applied in machine learning for diverse purposes, including improving image segmentation (Barbieri et al., 2011; Li et al., 2020), reducing neural network depth (Quétu et al., 2024), interpreting model behavior (Gabrié et al., 2018), facilitating unsupervised learning (Li et al., 2022), and estimating binary configuration entropy via classifiers (Janik, 2019). However, to the best of our knowledge, entropy has not been used to evaluate or compare medical image segmentation models, nor to estimate their optimal training dataset size. In this work, we apply entropy to develop a functional form that predicts segmentation model performance. We now describe the procedure for developing the functional model.

### 3.1 TRAINING ML MODELS

Given an architecture for developing an ML model and a training dataset, we train three or more ML models using subsets of the training examples. The reason for using three or more models will become clear from the discussion below. Each ML model is trained with a different number of examples. It is important to ensure that the distribution of examples in each subset closely matches the distribution of the original dataset. To compute the three parameters in Eq. (4, 5), at least three ML models must be trained; however, using more models improves the approximation of the parameters. In the work reported here, we use five ML models.

Recall that the training dataset $\mathcal{D}^{(ti)}$ contains $ti$ examples, that is, $|\mathcal{D}^{(ti)}| = ti$. Let $\mathcal{D}^{(t1)}, \cdots, \mathcal{D}^{(tj)}$ be $j$ subset of the full training dataset $\mathcal{D}_t$ where $t1 < t2, \cdots, t_{j-1} < tj < |\mathcal{D}_t|$. Let $\mathrm{ML}^{(ti)}$ denote the ML model trained using the training set $\mathcal{D}^{(ti)}$.

### 3.2 COMPUTING ENTROPY OF ML MODELS

All discussions in this section assume that the architecture and number of parameters used to develop an ML model remain unchanged. One ML model may differ from another only in the values of its trained parameters. Although different initializations may lead to different parameter values, the performance difference is typically insignificant if both models are trained on the same dataset. In contrast, increasing the size of the training dataset generally leads to a significant improvement in model performance.

The accuracy of a medical image segmentation ML model is typically measured using IoU scores or Intersection over Union (IoU) scores. Without loss of generality, the discussion here focuses on IoU

scores, although it applies equally to other similar scores. We begin by transforming the observed IoU scores into a discrete probability distribution.

**Probability distribution from IoU scores**  For a model $ML^{(ti)}$, we compute the IoU scores, $IoU^{(ti)}$, using the evaluation dataset $\mathcal{D}_e$ yielding $|\mathcal{D}_e|$ IoU scores. These IoU scores are divided into $k$ ranges (see Section 2.1 for details). Let $f_l^{(ti)}$ be the number of IoU scores falling within range $r_l$ for $1 \leq l \leq k$. Each $f_l^{(ti)}$ is divided by $|\mathcal{D}_e|$ to obtain the corresponding probability $p_l^{(ti)}$. The resulting probability distribution is denoted by $p^{(ti)} = \{p_1^{(ti)}, \cdots, p_k^{(ti)}\}$. Clearly, we have

$$p_l^{(ti)} = f_l^{(ti)}/|\mathcal{D}_e| \text{ and } \sum_{i=1}^{k} p_i^{(ti)} = 1. \tag{1}$$

An algorithmic outline for computing the probability distribution is shown in Appendix C

**Entropy of the trained ML models**  The probability distributions obtained from the Algorithm in Figure 7 are used to compute the entropy of the value of a trained ML model.

$$H(p^{(ti)}) = -\sum_{l=1}^{k} p_l^{(ti)} \log p_l^{(ti)} \tag{2}$$

**Fact about entropy**  It is well known that entropy is invariant under permutation of the components of a probability distribution. However, the probability distributions we use are derived from the IoU scores of corresponding ML models. Reordering the probability values is not permissible, as it would alter the estimated IoU score. For example, if $p_5 > p_9$ and we switch their positions, the resulting IoU score would increase, even though the entropy remains unchanged. Therefore, *when we are computing the entropy of an ML model with a probability distribution obtained from the IoU scores of the ML model, the entropy reflects the performance of the ML model.* We state this as Lemma 1 after defining *IoU score induced probability distribution*.

**Definition 3.** (**IoU score induced probability distribution**) Let the average IoU score $IoU_{\mathcal{D}_e}$ of an ML model be evaluated with a dataset $\mathcal{D}_e$. Let $p = \{p_i | i \in [k]\}$ be the $IoU_{\mathcal{D}_e}$ induced probability distribution, such that $p_i$ is the probability of the IoU score $IoU_i$. Then we have:

$$IoU_{\mathcal{D}_e} = \sum_{i \in [k]} p_i IoU_i \tag{3}$$

**Lemma 1.** (**IoU score preserving ML model entropy**) Let $p = \{p_i | i \in [k]\}$ be the IoU score $IoU_{\mathcal{D}_e}$ induced probability distribution of a ML model. Then the IoU score preserving entropy $H(P) = -\sum_{i \in [k]} p_i \log p_i$ of the ML model is unique.

*Proof.* The proof of the Lemma 1 follows from the discussion just before Definition 3. □

As the training dataset size is increased, the average IoU scores of the ML models increase because the frequency of higher IoU scores shifts to the right; the shift of the IoU scores to the right makes the probability distribution more skewed towards the right. Consequently, as the training dataset size increases, the entropy of the ML model decreases at a *negative* exponential rate.

We used the $j = 9$ ML models, each trained with a different training dataset size, to calculate the entropy values and estimate the parameters of our functional form.

### 3.3  POTENTIAL FUNCTIONAL FORMS FOR PERFORMANCE ESTIMATION

We denote the entropy of an ML model trained with $n$ training examples as $H(n)$ or $H(p^{(n)})$. We investigate two classes of functional forms:

$$\text{(i) Exponential decay:} \quad y(n) = ae^{-bn} + c \tag{4}$$

$$\text{(ii) Power-law decay:} \quad y(n) = an^{-b} + c \tag{5}$$

Let $n > 0$ denote the training dataset size, and $y(n)$ the corresponding model entropy. In both 4 and 5, $b$ is the entropy decay rate (larger $b$ means faster entropy decrease), $c$ is the asymptotic entropy (the value approaches to when dataset size approaches to infinity), and the maximum entropy is $(a + c)$. For a good estimation of values 3 parameters we need at least 6 or more empirical entropy values for each of the ML models. Next we provide a method for empirically estimating parameters $a$, $b$, and $c$ of the functional forms assuming that we have 8 or more entropy values.

**An algorithmic outline for the functional form development** All major steps for developing a functional form from IoU scores of $j$ ML models are summarized in Figure 2.

```
PROCEDURE COMPUTE_MODEL_PARAMETERS()
01 Begin
02    For i = 1 to j                                            {j ≥ 8}
03       Training a model ML⁽ᵗⁱ⁾ with the training dataset D⁽ᵗⁱ⁾
04       Compute IoU scores IoU⁽ᵗⁱ⁾ of the ML⁽ᵗⁱ⁾ with Evaluation Dataset 𝒟ₑ
05       Convert IoU Scores to Probability p⁽ᵗⁱ⁾              { Algorithm in Figure 7}
06       Compute Entropy of the Model H(p⁽ᵗⁱ⁾)                 {Equation (2)}
07    Compute model parameters a, b, c from H(p⁽ᵗ¹⁾), ⋯ , H(p⁽ᵗʲ⁾)
08    Plot the entropy values, H(p⁽ᵗ¹⁾), ⋯ , H(p⁽ᵗʲ⁾) against dataset size
09 End
```

Figure 2: Algorithm for computing and plotting dataset sizes vs entropy of ML models.

Our extensive evaluation of two functional forms revealed that exponential decay (Eq. 4) is a better functional form than the power law (Eq. 5). The results for exponential decay is shown in the Section 5 and that for the other functional form is shown in the Appendix A.

In our empirical study for the exponential decay functional form, reported in Section 5, on the Kvasir dataset (Jha et al., 2020), U-Net (Ronneberger et al., 2015) showed a higher $c$ than SegFormer (Xie et al., 2021) and PolypPVT (Dong et al., 2021), implying lower asymptotic accuracy. Figure 1 compares the predicted performances of these architectures; all had similar parameter counts. See Section 5 for details.

## 4 APPLICATIONS OF THE FUNCTIONAL FORM

**Comparing different architectures:** Models for each architecture are trained using the method in Sections 3.1–3.3, and one functional form is fitted per architecture. For a fair comparison of two models should have same number of parameters, but most cases that is impossible and models with similar number for parameter are compared. To compare the models, the functional forms of the corresponding models are plotted against training dataset size (see Fig. 1), or values of the constants $a$, $b$, and $c$ can be compared.

**Comparing effect of number of parameters:** The same steps as above are followed, except the number of parameters among models of the same architecture varies (see Fig. 4).

**Asymptotic accuracy estimation:** In Equations 4 and 5, $c$ is the asymptotic value of the functional forms. It informs us that a given architecture will not reach zero entropy even with extremely large number of training examples, which is a limitation of the architecture.

**Estimating cost-optimal training dataset size:** Eq. 6 for determining the optimal dataset size obtained using the functional form 4 is derived in the next section. Similar equation can be obtained for the functional form Eq. 5.

### 4.1 DATASET SIZE ESTIMATION FOR COST-OPTIMAL ML MODEL DEVELOPMENT

In this section, we propose a method for estimating the training dataset size required to develop a cost-optimal ML model. An ML model is said to be cost-optimal when the cost of gathering and annotating additional training data equals the reward gained from the resulting improvement in the model's performance. The performance gain from additional training data will eventually be

outweighed by the cost of data gathering. Equation 6 in Lemma 2 provides a formula for determining cost-optimal training dataset size.

**Lemma 2.** (Cost-optimal training dataset size) Give the functional form $y(n) = ae^{-bn} + c$ for a ML model, and the cost of acquiring an annotated training example $\alpha$, the cost-optimal training dataset size is (a proof is given in Appendix B):

$$n_{cost-optimal} = -\frac{1}{b} \log \frac{\alpha}{ab}. \tag{6}$$

Once we have an optimal dataset size $n_{cost-optimal}$, we can find the entropy value for this $n$ from the functional form's equation 4. To estimate the accuracy of the ML model from its entropy value, we need to revert the entropy value to an IoU score. Next, we describe a heuristic for the conversion.

### 4.2 ENTROPY TO PERFORMANCE ESTIMATION

Recall that to compute entropy, IoU scores were converted into a probability distribution, and entropy was then evaluated from this distribution. Since both steps are irreversible, recovering a unique probability distribution (and thus estimating performance) from entropy alone is not possible. To address this, we propose a heuristic approach that leverages two constants: $(i)$ the normalization condition that the sum of probabilities is 1, and $(ii)$ the entropy value for a given training dataset size obtained from the functional form.

#### 4.2.1 ESTIMATION OF PROBABILITY DISTRIBUTION

The underlying probability distribution of IoU scores is unknown. Empirically, as training size increases, the average IoU shifts toward higher values. To model this behavior, we approximate the distribution $p$ by a geometric progression with ratio $\rho$, which would capture the skewness of the mass toward higher IoU bins. However, for smaller mask areas models fail to segment the accurately; same is the situation for larger mask areas. This situation is illustrated in Fig. 3.

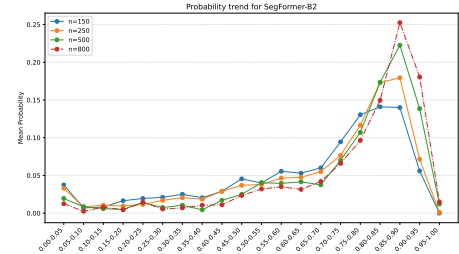

Figure 3: Experimental probability distributions of IoU scores with dataset sizes 150, 250, 500, and 800 for SegFormer (B2). While shift toward right is clear but needs some correction factor.

To reconcile this discrepancy while preserving the geometric structure of the main body of the distribution, we introduce a residual mass parameter $0 \le d < 1$. Conceptually, we assume that geometric distribution capture probability value (1-d). Equation 7 formally captures this assumption.

Formally, the distribution is given by

$$p_1 + p_1\rho + \cdots + p_1\rho^{k-1} = p_1 \frac{\rho^k - 1}{\rho - 1} = 1 - d, \tag{7}$$

which yields

$$p_i = (1 - d)\frac{\rho - 1}{\rho^k - 1}\rho^{i-1}, \quad i = 1, \ldots, k, \qquad p_{k+1} = d. \tag{8}$$

It is important to note that value of $\rho > 1$ and increases as number of training examples increases.

To estimate $\rho$, for a given training dataset size $tj$ we equate the entropy of the model from functional form for $tj$. Denoting the entropy of the empirical distribution as $H(p^{(tj)})$, we have

$$H(\rho, d) = -\sum_{i=1}^{k} p_i \log p_i - d \log d. \tag{9}$$

After substituting values of $p_i$ from Eq. equation 8 in Eq. 9 and performing some algebraic manipulation, a closed-form expression for $H(\rho, d)$ can be found and is stated as a Lemma next.

**Lemma 3** (Closed-form entropy expression). Denoting $(\rho^k - 1)/(\rho - 1)$ by $S$, a close-form solution of Eq. 9 is given by,

$$H(\rho, d) = (1 - d)\left[\log(S) - \frac{\rho \log(\rho)}{\rho^{k-1}}\left(k\rho^{k-1} - S\right)\right] - (1 - d)\log(1 - d) - d\log d. \quad (10)$$

A complete proof of Lemma 3 is provided in Appendix D. For each training size $tj$, we compute $H(\rho, d)$ from the functional form, solve Eq. 10 numerically for $\rho$, and recover the probability distribution $p^{(tj)}$. This estimated distribution is then used to compute the expected IoU score.

### 4.2.2 ML MODEL'S ACCURACY ESTIMATION

Solving Equation (10) for $\rho$ and $d$, and substituting into Equation (8), yields $k$ probabilities $\{p_i\}_{i=1}^k$. These are used to estimate the model's IoU for a given training dataset size:

$$\hat{\text{IoU}} = \sum_{i=1}^k p_i \hat{\text{IoU}}_i. \quad (11)$$

For each bin $r_i = ((i - 1)/k, \, i/k]$, three choices are possible:

$$\hat{\text{IoU}}_i^{\min} = \frac{i-1}{k}, \quad \hat{\text{IoU}}_i^{\mathrm{mid}} = \frac{2i-1}{2k}, \quad \hat{\text{IoU}}_i^{\max} = \frac{i}{k}.$$

The midpoint is generally the most practical estimate, while the left and right bounds provide lower and upper limits. As $k$ increases, the interval width $1/k$ decreases, tightening the estimates.

## 5 EVALUATION OF PROPOSED METHOD: CASE STUDIES

### 5.1 DATASET AND MODEL ARCHITECTURES

We use the Kvasir-Seg dataset (Jha et al., 2020), which contains 1,000 examples, each consisting of an image and its annotated binary mask. The dataset is partitioned into an evaluation set of 200 examples, denoted as $\mathcal{D}_e$, and 800 examples used for training ML models.

For empirical evaluation, we used U-Net (Ronneberger et al., 2015), SegFormer (Xie et al., 2021), and Polyp-PVT (Dong et al., 2021). U-Net is a CNN with an encoder–decoder design and skip connections, widely used in biomedical segmentation. SegFormer is a transformer-based model combining hierarchical encoders with lightweight decoders, while Polyp-PVT uses a Pyramid Vision Transformer backbone tailored for polyp segmentation. Table 1 summarizes the model variants, including parameter counts (M) and FLOPs (B). We evaluated three SegFormer variants (B0, B1, B3) of different sizes. Figure 4 shows that, even with the same training data, their performances vary with parameter count. To isolate architectural effects, U-Net(86) and Polyp-PVT(B3) were chosen to match the parameter size of SegFormer B2 (Fig. 1). Each model was trained ten times, and results were averaged for stability.

Table 1: Model Variants, Parameters, and FLOPs Comparison

| Model | Params (M) | FLOPs (G) |
|---|---|---|
| SegFormer(B0) | 7.72 | 3.30 |
| SegFormer(B1) | 29.68 | 9.67 |
| SegFormer(B2) | 56.41 | 24.83 |
| Polyp-PVT(B3) | 44.98 | 9.02 |
| U-Net(86) | 56.04 | 98.76 |

Table 2: Constants for values $a$, $b$, and $c$ for functional form $y = ae^{-bn} + c$. Plots are in Figs. 1 and 4.

| Model | a | b | c |
|---|---|---|---|
| SegFormer-B0 | 0.695 | 0.201 | 2.248 |
| SegFormer-B1 | 0.873 | 0.117 | 2.005 |
| SegFormer-B2 | 0.664 | 0.332 | 2.272 |
| PolypPVT-B3 | 0.787 | 0.434 | 2.056 |
| U-Net (86) | 0.520 | 0.275 | 2.397 |

### 5.2 EXPERIMENTAL SETUP

**Computing Resources** All training experiments were performed on two workstations, each with AMD Ryzen Threadripper PRO 5945WX 12-Cores with 128 GB memory, and NVIDIA RTX A4500 GPU with 20 GB of VRAM, providing sufficient computational resources for training models of varying complexities.

**Hyperparameter Settings** From extensive evaluation, we found that SegFormer, PolypPVT, and U-Net models performed well with the following hyperparameter values: (*i*) Loss Function: CrossEntropy Loss (*ii*) Optimizer: AdamW (*iii*) convergence value $1 \times 10^{-4}$, (*iv*) batch size 12, (*v*) moving average window size 10, and (*vi*) learning rate $1 \times 10^{-3}$.

## 5.3 FUNCTIONAL FORMS

Recall that to distinguish between the functional forms developed in this work and the image segmentation models trained for the study, the latter are called ML (machine learning) models.

For developing a functional form, we need the performance of several ML models, each trained with a different training dataset size. For ML model training, we further partitioned the 800 examples into different set sizes. To be more specific, we develop ML models with dataset sizes of 100, 150, 200, 250, 300, 400, 500, 600, 700, and 800. We used IoU scores from the first 9 models for estimating parameters $a$, $b$, and $c$ for both the functional forms in Eq. 4, 5. Parameter values of the five image segmentation architecture variants in Table 1 are shown in Table 2.

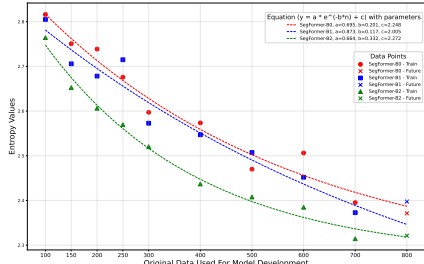

Figure 4: Plots showing functional forms for SegFormer B0, B1 and B2. The entropy of 9 of the 10 trained ML models was used to fit the functional form $y = ae^{-bn} + c$

**Different architectures with approximately equal number of parameters** After training each ML model, we obtain the Intersection of Union score (denoted as IoU) for all the examples in the evaluation dataset. These IoU scores are grouped into 20 equal-sized bins $b_1, b_2, \cdots, b_{20}$; all IoU scores between 0 to 0.05 are placed in the bin $b_1$, all IoU scores between 0.05 to 0.1 are placed in the bin $b_2$, and so on. The IoU scores in each bin are normalized to the probabilities, that is, the number of scores in bin $b_i$ is divided by 200 to obtain $p_i$, $i \in [20]$. Plots in Fig. 1 for functional forms of U-Net 86, SegFormer B2, and PolypPVT B3 give a visual comparison of their performances. From the plots, it is easy to see that PolypPVT is the best architecture and UNet is the worst.

**Same architecture but different number of parameters** Plots in Figure 4 show 3 plots for the SegFormer architecture with different numbers of parameters. It is evident that our functional forms can illustrate the fact that more parameters create a better ML model.

**Comparison between both functional forms** We compared the exponential decay form (Eq. 4, Table 2, Figs. 1, 4) with the power law form (Eq. 5, Table 4, Figs. 5, 6). Power law fits are numerically stable for small dataset sizes, but their asymptotes are consistently low, implying unrealistically high confidence as $n \to \infty$. In contrast, exponential decay is more sensitive to architectural differences but converges toward empirically plausible plateaus, consistent with the finite irreducible uncertainty of the task.

## 5.4 IOU SCORE ESTIMATE FOR A GIVEN DATASET SIZE

Table 3 reports observed IoU scores for five ML models trained with 800 examples, along with entropy values computed from 20 equal-sized bins and the corresponding asymptotic IoU ranges. To obtain these estimates, we first fit the functional form to derive $\rho$ and $d$, then computed probabilities using Equations 8–10, and finally applied Equation 11 to calculate minimum, maximum, and mean asymptotic IoU values. Comparing the observed IoUs (first column) with the estimated ranges (last three columns), we find close agreement, indicating that the procedure is reliable. The same method can be applied to estimate the IoU of a model trained with any dataset size by first computing the entropy value from Eq. 4.

For completeness, we also repeated the same process using the power law form (Table 5). A comparison of Tables 3 and 5 shows that exponential decay achieves closer agreement with observed IoUs and requires only negligible corrections ($d \leq 0.13$). In contrast, the power law form requires substantially larger corrections (up to $d = 0.15$) to reconcile predictions with the data, revealing a

Table 3: Observed IoU, entropy, and estimated ranges (with/without $d$ correction) for functional form $y = ae^{-bn} + c$.

| Model | Observed | | Estimated ($d = 0$) | | | | Estimated ($d > 0$) | | | |
|---|---|---|---|---|---|---|---|---|---|---|
| | IoU | $H$ | $d$ | $\text{IoU}_{\min}$ | $\text{IoU}_{\max}$ | $\text{IoU}_{\text{mean}}$ | $d$ | $\text{IoU}_{\min}$ | $\text{IoU}_{\max}$ | $\text{IoU}_{\text{mean}}$ |
| **SegFormer-B0** | 0.757 | 2.248 | 0.0 | 0.800 | 0.850 | 0.825 | 0.08 | 0.752 | 0.798 | 0.775 |
| **SegFormer-B1** | 0.747 | 2.005 | 0.0 | 0.838 | 0.888 | 0.863 | 0.13 | 0.745 | 0.788 | 0.766 |
| **SegFormer-B2** | 0.763 | 2.272 | 0.0 | 0.795 | 0.845 | 0.820 | 0.07 | 0.756 | 0.803 | 0.779 |
| **PolypPVT-B3** | 0.835 | 2.056 | 0.0 | 0.830 | 0.880 | 0.855 | 0.01 | 0.827 | 0.877 | 0.852 |
| **U-Net (86)** | 0.776 | 2.397 | 0.0 | 0.771 | 0.821 | 0.796 | 0.01 | 0.770 | 0.819 | 0.794 |

systematic bias. These results confirm that the exponential decay functional form provides the most consistent and theoretically justified characterization of model performance.

**Cost-optimal dataset sizes and corresponding IoU scores**     Once the cost of data collection and reward for IoU score improvement is known, Equation 6 can be used to calculate the cost-optimal value of training dataset size, which can then be used to obtain the entropy value from the functional form. The following steps describe how one can compute cost-optimal IoU scores. Because we neither have data collection costs nor rewards for performance gain, we did not create hypothetical data.

## 6  CONCLUSION AND FUTURE WORK

In this paper, we have proposed and developed a method for creating functional forms for ML models used to segment medical images. The proposed method develops several ML models, each with a different number of training examples, and estimates the entropy of these models. Because larger training datasets create better ML models, the entropy of the trained models decreases as a function of the number of training examples. We evaluate two classes of functional forms, exponential decay and power law, for a given several architectures and training dataset. WE show that functional forms are useful for comparing the performance of ML models developed with different architectures, but with the same training dataset. It is also useful for comparing the performance of different ML models developed using different variants of the same architecture. It can also be used to estimate the performance of an ML model that could be trained with any training example set size, without training an actual ML model; the training dataset size could be larger than what is *really* available. It can even be used to calculate the cost-optimal training dataset size. Further details can be found in Sections 4, 5, and Appendix A.

We have empirically evaluated functional forms developed using the method we have proposed here. We used Kvasir (Jha et al., 2020) dataset and three image segmentation architecture: UNet (Ronneberger et al., 2015), SegFormer (Xie et al., 2021), and ploypPVT (Dong et al., 2021). Our evaluations show that the assumptions we made for estimating IoU scores from the entropy obtained from functional form are valid. We plan to continue to validate our method for larger ML models and more datasets in the near future. However, we are constrained by available computing resources. For the results we have reported here, we have used an average of three ML models developed with different initializations. We could not compare our work with any previous work, because to the best of our knowledge, there is no method for developing functional forms from ML models.

**Limitations**     Our method develops functional forms from image segmentation models by assuming that evaluation scores can be converted into probability distributions. We further assume that the distribution of a well-trained model can be approximated with a geometric sequence plus one more term, and that entropy can be approximately reverted to recover this distribution. Empirical evaluation confirmed this to be a good approximation.

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

## APPENDICES

## A  DATA FOR POWER-LAW FUNCTIONAL FORM

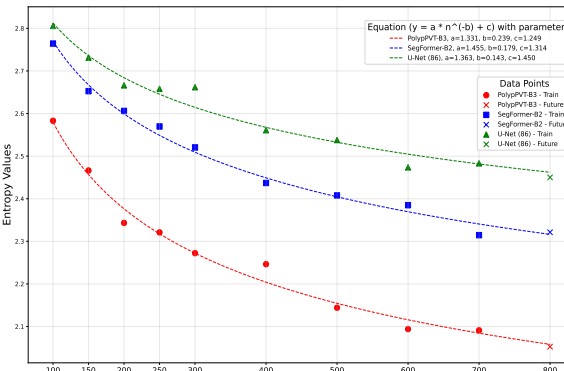

Figure 5: functional forms for U-Net, SegFormer, and PolypPVT. The entropy of 9 of the 10 trained ML models was used to fit the functional model.

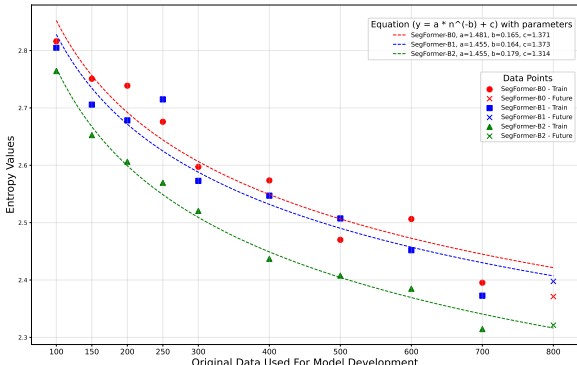

Figure 6: functional forms for SegFormer B0, B1, B2. The entropy of 9 of the 10 trained ML models was used to fit the functional model.

Table 4: Constants for values a, b, and c for functional form $(y(n) = an^{-b} + c)$. Plots are in Figs. 5 and 4.

| Model | a | b | c |
|---|---|---|---|
| SegFormer-B0 | 1.481 | 0.165 | 1.371 |
| SegFormer-B1 | 1.455 | 0.164 | 1.373 |
| SegFormer-B2 | 1.455 | 0.179 | 1.314 |
| PolypPVT-B3 | 1.331 | 0.239 | 1.249 |
| U-Net (86) | 1.363 | 0.143 | 1.450 |

Table 5: Observed IoU, entropy, and estimated ranges (with/without $d$ correction) for functional form $y = an^{-b} + c$

| Model | Observed | | Estimated ($d = 0$) | | | | Estimated ($d > 0$) | | | |
|---|---|---|---|---|---|---|---|---|---|---|
| | IoU | $H$ | $d$ | $\text{IoU}_{min}$ | $\text{IoU}_{max}$ | $\text{IoU}_{mean}$ | $d$ | $\text{IoU}_{min}$ | $\text{IoU}_{max}$ | $\text{IoU}_{mean}$ |
| **SegFormer-B0** | 0.757 | 1.371 | 0.0 | 0.856 | 0.906 | 0.881 | 0.14 | 0.752 | 0.795 | 0.773 |
| **SegFormer-B1** | 0.747 | 1.373 | 0.0 | 0.856 | 0.906 | 0.881 | 0.15 | 0.743 | 0.786 | 0.764 |
| **SegFormer-B2** | 0.763 | 1.314 | 0.0 | 0.868 | 0.918 | 0.893 | 0.14 | 0.762 | 0.805 | 0.783 |
| **PolypPVT-B3** | 0.835 | 1.249 | 0.0 | 0.890 | 0.940 | 0.915 | 0.08 | 0.831 | 0.877 | 0.854 |
| **U-Net (86)** | 0.776 | 1.450 | 0.0 | 0.842 | 0.892 | 0.867 | 0.1 | 0.773 | 0.818 | 0.796 |

## B  PROOF OF LEMMA 2

For ease of reading the Lemma is stated first.

**Lemma 2.** (Cost-optimal training dataset size) Give a functional form $y(x) = ae^{-bx} + c$ for a ML model, and cost of acquiring an annotated a training example $\alpha$, the cost-optimal training dataset size is:

$$x = -\frac{1}{b} \log \frac{\alpha}{ab}. \tag{12}$$

*Proof.* As discussed in Section 3.3, the maximum entropy for the ML model is $(a + c)$. After training an ML model with a training dataset of size $n$, performance gain $G(n)$ of the ML model is given by:

$$G(n) = (a + c) - y(n) = (a + c) - (ae^{-bn} + c) = a - ae^{-bn} = a(1 - e^{-bn}). \tag{13}$$

Note that we have used right side of the Equation (4) to replace $y(n)$ in Equation (13). Assuming the cost of acquiring and annotating a training example is denoted by $\alpha$, the net performance gain (or loss), $G_{net}(x)$ is given by:

$$G_{net}(n) = G(n) - \alpha n = a(1 - e^{-bn}) - \alpha n = a - ae^{-bn} - \alpha n. \tag{14}$$

To find the optimal value of net performance gain, we take derivative of $G_{net}(n)$ with respect to $n$, set it equal to zero, and solve the obtained equation for $n$.

$$G'_{net}(n) = abe^{-bn} - \alpha = 0 \tag{15}$$

Solving Equation (15) for $n$, we get optimal value of the training dataset size:

$$n_{cost-optimal} = -\frac{1}{b} \log \frac{\alpha}{ab}. \tag{16}$$

$\square$

## C  ALGORITHM FOR COMPUTING PROBABILITY DISTRIBUTION FROM IOU SCORES

An algorithmic outline for computing the probability distribution is outlined below.

```
PROCEDURE COMPUTE_PROBABILITY_DISTRIBUTION(IOU^(ti))
01  Begin
02      For l = 1 to k
03          Let f_l^(ti) be the number of IoU score in the range r_l
04          p_l^(ti) = f_l^(ti)/|D_e|
05      p^(ti) = {p_1^(ti), ···, p_k^(ti)}
06  End
```

Figure 7: Algorithm for computing probability distribution for the model $ML^{(ti)}$.

## D  PROOF OF LEMMA 3

We restate Lemma 3 before proving it.

**Lemma 3.** (Closed-form expression for computing $\rho$) A closed form expression for computing $\rho$ is given by,

$$H(\rho) = (1 - d) \left[ \log(S) - \frac{\rho \log(\rho)}{\rho^{k-1}} \left( k\rho^{k-1} - S \right) \right] - (1 - d) \log(1 - d) - d \log(d). \tag{17}$$

*Proof.* Assume the first $k$ bins of the probability distribution follow a geometric progression with ratio $\rho$, while a hypothetical $(k+1)$-st bin carries residual mass $d$. Then for $i = 1, \ldots, k$ we have

$$p_i = (1 - d) \frac{\rho - 1}{\rho^k - 1} \rho^{i-1}, \qquad p_{k+1} = d. \tag{18}$$

Assuming probability distribution $p = \{p_i | i \in [n]$ is a geometric progression sequence with a ratio $\rho$, we have:

$$p_i = (1 - d)\frac{\rho - 1}{\rho^n - 1}\rho^{i-1} \quad \text{for } i \in [n]. \tag{19}$$

We know that, in case of geometric series,

$$S = \sum_{j=0}^{k-1} \rho^j = \frac{\rho^k - 1}{\rho - 1} \tag{20}$$

Now, if we normalize all the bins and take sum, it should sum to 1

$$q_i = \frac{p_i}{1 - d} \implies p_i = q_i(1 - d) \quad \text{and} \quad \sum_{i=1}^{k} q_i = 1 \tag{21}$$

Now from 7, We know:

$$p_i = (1 - d)\frac{\rho - 1}{\rho^n - 1}\rho^{i-1}$$

$$\implies \frac{p_i}{1 - d} = \frac{\rho - 1}{\rho^n - 1}\rho^{i-1} \tag{22}$$

$$\implies \frac{p_i}{1 - d} = \frac{\rho^{i-1}}{S}$$

Thus, from eq 21 and eq 22, we get,

$$q_i = \frac{p_i}{1 - d} = \frac{\rho^{i-1}}{S} \tag{23}$$

At this point, we have total (k+1) bin distributions; Mass $(1 - d)$ is spread across $k$ bins and one single left over bin with mass $d$. Thus,

$$H(\rho, d) = -\sum_{i=1}^{k} p_i \log p_i - d \log d$$

$$\implies -\sum_{i=1}^{k} q_i(1 - d) \log\{(1 - d)q_i\} - d \log d$$

$$\implies -\sum_{i=1}^{k} q_i(1 - d)\{\log(1 - d) + \log q_i\} - d \log d \tag{24}$$

$$\implies -(1 - d)\log(1 - d)\sum_{i=1}^{k} q_i - (1 - d)\sum_{i=1}^{k} q_i \log q_i - d \log d$$

$$\implies -(1 - d)\sum_{i=1}^{k} q_i \log q_i - (1 - d)\log(1 - d) - d \log d$$

As we know from eq 23,

$$q_i = \frac{\rho^{i-1}}{S} \tag{25}$$
$$and, \quad \log q_i = (i - 1)\log \rho - \log S$$

Thus,

$$-\sum_{i=1}^{k} q_i \log q_i = -\sum_{i=1}^{k} q_i \frac{\rho^{i-1}}{S}\{(i-1)\log \rho - \log S\}$$

$$-\sum_{i=1}^{k} q_i \log q_i = -\sum_{j=0}^{k-1} q_i \frac{\rho^{j}}{S}\{j \log \rho - \log S\}$$

$$-\sum_{i=1}^{k} q_i \log q_i = -\frac{\log \rho}{S} \sum_{j=0}^{k-1} j\rho^j + \frac{\log S}{S} \sum_{j=0}^{k-1} \rho^j \tag{26}$$

$$-\sum_{i=1}^{k} q_i \log q_i = -\frac{\log \rho}{S} \sum_{j=0}^{k-1} j\rho^j + \frac{\log S}{S} S$$

$$-\sum_{i=1}^{k} q_i \log q_i = \log S - \frac{\log \rho}{S} \sum_{j=0}^{k-1} j\rho^j$$

Now, we take differentiation of $\rho$ for both side of eq 20, we have

$$\frac{\partial}{\partial \rho}\left(\sum_{j=0}^{k-1} \rho^j\right) = \frac{\partial}{\partial \rho}\left(\frac{\rho^k - 1}{\rho - 1}\right)$$

$$\implies \sum_{i=j}^{k-1} j\rho^{j-1} = \frac{(k\rho^{k-1})(\rho - 1) - (\rho^k - 1)}{(\rho - 1)^2}$$

$$\implies \frac{\sum_{i=j}^{k-1} j\rho^j}{\rho} = \frac{(k\rho^{k-1})(\rho - 1) - (\rho^k - 1)}{(\rho - 1)^2} \tag{27}$$

$$\implies \sum_{i=j}^{k-1} j\rho^j = \rho\frac{(k\rho^{k-1})(\rho - 1) - (\rho^k - 1)}{(\rho - 1)^2}$$

Using eq 27 in eq 25 we get,

$$-\sum_{i=1}^{k} q_i \log q_i = \log S - \frac{\rho \log \rho}{S} \frac{(k\rho^{k-1})(\rho - 1) - (\rho^k - 1)}{(\rho - 1)^2}$$

$$-\sum_{i=1}^{k} q_i \log q_i = \log S - \frac{\rho \log \rho(k\rho^{k-1} - S)}{\rho^k - 1} \tag{28}$$

Thus our final closed form equation 24 becomes,

$$H(\rho, d) = (1 - d)\left[\log(S) - \frac{\rho \log(\rho)}{\rho^{k-1}}\left(k\rho^{k-1} - S\right)\right] - (1 - d)\log(1 - d) - d\log(d). \tag{29}$$

$\square$

