# OpenReview forum: "Functional Form For Segmentation Accuracy Prediction for Comparing Architectures and Estimating Cost-Optimal Training Dataset"
_ICLR.cc/2026/Conference — ICLR 2026 Conference Withdrawn Submission_

### Official Review · Reviewer_Gici · 2025-10-21

**Soundness:** 2
**Presentation:** 2
**Contribution:** 3
**Rating:** 4
**Confidence:** 3

**Summary:**

The paper proposes to model segmentation “performance” via the **entropy of the IoU histogram** computed on a held-out set, and to fit **functional forms** of entropy vs. training-set size—exponential (y(n)=ae^{-bn}+c) and power-law (y(n)=an^{-b}+c)—to extrapolate asymptotic behavior, compare architectures, and derive a **cost-optimal dataset size** (Lemma 2). It then introduces a heuristic to **invert entropy back to IoU** by assuming a geometric probability shape plus a residual mass (d). The method is evaluated on **Kvasir-Seg (1,000 images)** with U-Net, SegFormer, and Polyp-PVT variants.

**Strengths:**

* Clear, end-to-end **operational pipeline**: IoU → histogram → entropy → functional fit; plus a practical inversion heuristic for estimating IoU from entropy.
* The **case study** is well-scoped and provides ablations across model variants and dataset sizes (100–800).

**Weaknesses:**

# Major Concerns

1. **“Entropy” as a performance proxy lacks theoretical grounding beyond plausibility.**
   The paper defines entropy over the IoU-induced histogram and notes that higher training size tends to “shift mass rightward” (higher IoU) and **decrease entropy**. However, **no monotonic, one-to-one link** is established between entropy and mean IoU; many distinct distributions can share the same entropy yet have different means, which the paper itself implicitly recognizes when introducing a heuristic inversion. I recommend: (i) empirical monotonicity checks between entropy and mean IoU across different bin counts (k) and binning schemes; (ii) correlations (with CIs) between entropy and standard metrics (mIoU/Dice/clDice) across seeds; (iii) sensitivity to (k) and to evaluation-set size. These additions would turn the current qualitative claim (“entropy reflects performance”) into a quantitatively supported statement.

2. **Lemma 2’s “cost-optimal dataset size” lacks units/operational meaning in its current form.**
   Lemma 2 optimizes **net gain** (G_{\text{net}}(n)=a(1-e^{-bn})-\alpha n) and yields (n^*=-\frac{1}{b}\log!\frac{\alpha}{ab}). But all derivations are in the **entropy space** (nats), so (G'(n)=ab e^{-bn}) has units of **nats/sample**; consequently (\alpha) must also be **nats/sample**, not dollars or minutes. Please either (A) explicitly frame (\alpha) as an **entropy-threshold stopping rule** (and study its sensitivity), or (B) provide a **value mapping** from entropy changes to IoU (via your Sec. 4.2 heuristic) and then to money/time (e.g., $/%IoU or minutes/%IoU), so that the economic (n^*) becomes reproducible for practitioners. A worked example with concrete numbers would greatly help.

3. **External validity is weak; evidence is too narrow to support general claims.**
   All evidence comes from **one dataset and one task type (polyp segmentation)**, with a single evaluation protocol. To claim that exponential fits are generally preferable to power-law fits, I would expect: (i) multiple datasets across modalities (retina/CT/MRI/skin/organ), (ii) multiple metrics (mIoU, Dice, clDice), (iii) robustness to label noise/small masks, and (iv) error bars across seeds for fitted (a,b,c) and extrapolation error. As written, the “exponential vs. power law” conclusion feels over-generalized.

---

# Secondary Concerns (Presentation & Clarity)

* **Equation referencing**: where you write “4 and 5”, please use “**Eq. (4)** and **Eq. (5)**” consistently (the section introducing the two forms is around the Eq. (4)/(5) block).
* **Figure 1**: font is small and the caption is terse. Please enlarge axis/legend fonts and clarify what each curve represents and how entropy values were computed (bin count, log base). The link from Fig. 1 to Table 2/4 could also be stated.

**Questions:**

1. **Empirical validation of entropy as a proxy**: correlations and monotonicity tests vs. mIoU/Dice across varying (k) (e.g., 10/20/50) and evaluation sizes; sensitivity plots.
2. **Operationalizing Lemma 2**: either (i) report decisions under several **entropy-thresholds** (nats/sample) with ablation, or (ii) provide a **calibration curve** ( \Delta \text{IoU}) vs. (-\Delta H) and a simple $$ or minutes mapping to obtain a practical (n^*).
3. **Broaden the evaluation**: add 3–5 datasets beyond Kvasir-Seg and report confidence intervals for fitted parameters and extrapolated asymptotes; include noise/imbalance scenarios and topology-aware metrics where relevant.
4. **Clarify the inversion heuristic**: define how (d) is set; show stability of the solution of Eq. (10) for (\rho) (and (d)) and the effect of choosing lower/mid/upper bin representatives in Eq. (11).

---

### Official Review · Reviewer_bRFF · 2025-10-27

**Soundness:** 2
**Presentation:** 2
**Contribution:** 2
**Rating:** 2
**Confidence:** 4

**Summary:**

This paper proposes an approach to model how medical image segmentation neural network performance changes as a function of training dataset size. The authors propose a method of converting empirical performance metrics (observed from training neural networks with varying amounts of data) into entropy, then fitting a functional form to entropy vs. dataset size to predict future model performance. The authors validate their approach on one dataset with three segmentation models, using their approach to compare how different architectures and parameters counts scale with dataset size. The authors also propose extensions for computing the cost-optimal dataset size and converting the predicted entropy score back into an IoU score.

**Strengths:**

Underexplored but needed research area
- Due to the cost of labeling segmentation data, particularly in application areas like medicine that require expert labelers, the proposed aim of predicting how model performance will scale with additional training data is interesting and of high value. This is a relatively unexplored research area, and the proposed solution is general (e.g., model agnostic, dataset agnostic), making the work useful.

The functional form is interpretable and the proposed workflow is clearly presented, aiding in understanding and implementation.

Practical use cases of such a methodology are defined and explored, including:
- Using this tool to compare architectures.
- Using this tool to compare model sizes.
- Using this tool to predict a future model’s IoU.
- Using this tool to predict asymptotic IoU.

**Weaknesses:**

Unconvincing empirical evidence that the method is practically useful
- Only one dataset is used with three architectures to evaluate the proposed method. I find this amount of dataset/architecture pairs unconvincing to support the claim this is a generalizable, robust method for modeling performance as a function of dataset size.
- The results do not seem to be quantitatively validated. For example, no metrics assessing fit quality are provided, no assessment of average error, etc. The results are presented largely without narration or assessment.
- The authors report each model was trained ten times and results were averaged, yet there are no confidence intervals, standard deviations, etc. reported. Some of the empirical results deviate from their functional form (e.g., Figure 4, Table 3) significantly. The fact these empirical results are the average of 10 runs suggests the variation of empirical findings from functional form may be even larger (though I cannot assess this without confidence intervals, variance, etc.) again limiting the practicality/accuracy of the proposed approach.
- The current experiments use nine models trained with different amounts of labeled data to predict how a tenth model, trained with only a modest amount of more data, will behave. I find this setting impractical and not that useful: you are training many models (which is compute/time intensive) to predict how the performance of a model trained with only an incremental amount of more data will behave. I would find this method more convincing and practically useful if you could predict future performance accurately for larger data labeling deltas, and/or with fewer initial trained models.
- The analysis and motivation are all limited to medical image segmentation, although there is nothing about the method that is specific to medical imaging. The paper would be stronger if they validated the approach for general segmentation models.
- As far as I see, the predicted asymptotic performance limits (which I find to be a compelling use case) are not empirically/approximately confirmed.
- Only IoU is evaluated. The authors state all results apply to other “similar” performance metrics, but no support is provided for this statement.
- The cost-optimal dataset size procedure is proposed but not validated or supported with any evidence.

Unclear generalizability to common/practical settings
- How small of a dataset can you use to conduct these analyses? E.g. if I train a model with 1, 5, 10, and 20 samples, can I predict performance with 100, 200, 400 samples?
- How do common tools for enhancing model performance with limited labeled data (e.g., using data augmentation, using a pretrained model) impact the functional form accuracy?
- How do hyperparameters fit into this framework? E.g., learning rate, batch size, optimizer. Are these retuned for each new dataset size, or left the same? Does that impact the scaling results?
- How would this work with multiclass tasks?
- How does the choice of n=20 bins impact performance of the proposed methodology?

The writing should be made more precise
- In line 147 the authors state they use 5 ML models in this work, but in line 206 (and Figure 1, etc) it appears they use 8 ML models.
- In line 145-146 the authors state at minimum 3 ML models are required to estimate parameters, but in line 219 the authors state they need 6 or more observations, then provide an algorithm that requires 8 observations. It’s unclear how many models are needed to employ this method successfully.
- Many statements lack citations/evidence, a few examples below
1. “Machine learning (ML) models for medical image segmentation are reducing healthcare costs...” If it is true that segmentation models are reducing healthcare costs, this should be supported with a citation.
2. “Consequently, researchers have largely concentrated on architectural innovation.” A great deal of work has gone into training strategies that reduce labelling burden (self supervised, semi supervised, unsupervised, weakly supervised learning; data augmentation; foundation models; etc.)
- Grammar/clarity line 42, I think you mean “accuracy gain scales sub-linearly with dataset size”
- Equations are often referenced without “Eq.”, for example in line 262: “…using the functional form 4 is derived…”
- Full paper needs a pass for grammar and formatting errors.

**Questions:**

Addressed in weaknesses.

---

### Official Review · Reviewer_zrCz · 2025-11-01

**Soundness:** 3
**Presentation:** 3
**Contribution:** 3
**Rating:** 6
**Confidence:** 4

**Summary:**

This paper introduces a novel methodology for predicting the performance of medical image segmentation models as a function of training dataset size. The authors propose converting Intersection over Union (IoU) scores into entropy values and fitting these to functional forms, specifically exponential decay and power-law models, to estimate asymptotic performance, compare architectures, and determine cost-optimal dataset sizes. The method is empirically validated on the Kvasir-Seg dataset using U-Net, SegFormer, and Polyp-PVT architectures. The authors also provide a heuristic for converting entropy back to IoU scores and derive an analytical expression for cost-optimal dataset size.

**Strengths:**

1.The idea of using entropy derived from IoU distributions to model performance scaling is innovative and addresses an important gap in the literature. The application to medical image segmentation is timely and relevant, given the high cost of data annotation in this domain.
2.The paper provides a clear and systematic procedure for deriving functional forms, including detailed algorithms and proofs. The use of multiple models and dataset sizes for parameter estimation adds robustness to the approach.

**Weaknesses:**

1.The study is restricted to one dataset (Kvasir-Seg) and three architectures. Broader validation across multiple datasets (e.g., medical imaging benchmarks beyond polyps) and more diverse architectures would strengthen the claims.
2.The method for converting entropy back to IoU scores relies on a geometric distribution assumption and a residual mass parameter d. While effective, this step is heuristic and may not generalize well without further validation.
3.The functional forms assume a fixed model architecture and parameter count. The impact of architectural variations or hyperparameter tuning on the entropy-performance relationship is not explored.
4. The authors note that no prior work exists for direct comparison, but the paper would benefit from comparing against alternative scaling laws (e.g., from classification or language modeling) adapted to segmentation.
5. Some sections (e.g., entropy derivation, proof of Lemma 3) are dense and could be better explained for a broader audience. The figures and tables are informative but could be more clearly referenced in the main text. The writing occasionally lacks fluency, with minor grammatical errors and awkward phrasing.

**Questions:**

1.Validate the method on at least one additional public medical segmentation dataset.
2.Discuss the sensitivity of the functional form to architectural changes or hyper-parameters.
3.Compare the entropy-based scaling law with a simplified baseline (e.g., direct curve fitting to IoU vs. size).
4. Include a brief discussion on computational requirements and practical applicability.

---

### Official Review · Reviewer_M2fS · 2025-11-01

**Soundness:** 2
**Presentation:** 1
**Contribution:** 2
**Rating:** 2
**Confidence:** 4

**Summary:**

This paper proposes a framework for estimating the Intersection over Union (IoU) scores of medical image segmentation models as a function of training set size. The authors further develop a method to estimate the number of samples required to annotate in order to achieve a desired performance. The outlined method works by binning IoU scores of models, trained on data of varying size, on a validation set, and converting these into discrete probability distributions. The entropy of this distribution is then calculated, providing a measure of consistency in the IoU score on the validation set. Assuming that IoU scores increase as a function of dataset size and that this decrease in entropy, the authors fit a functional form to the observed entropy scores. Following this, a closed-form expression of this entropy is derived from which the discrete probability distribution of a model’s IoU scores can be recovered, enabling the calculation of the expected IoU score.

The developed framework allows practitioners to interpolate and extrapolate/interpolate expected segmentation accuracy, compare architectures even when only partial data is available, and analytically estimate the training dataset size that balances annotation cost with model performance.

**Strengths:**

**Motivation**: The problem addressed by the authors, namely, estimating the performance gains from increasing dataset sizes and a cost-optimal model, is timely and of practical relevance. The approach for estimating model performance is original and creative.

**Mathematical Formulation**: The development of the entropy-based performance estimation and the fitting of exponential/power-law decay models is mathematically sound.

**Validation**: Figures 1 and 3 provide visualizations that illustrate how entropy decays as the data size increases and how probability distributions shift, thereby strengthening the claim about model performance trends. The results in Table 3 show that the estimated IoU scores coincide with observed values, demonstrating the effectiveness of the approach.

**Weaknesses:**

1. The authors provide no code, which complicates the reproducibility of the reported results.

1. When estimating the optimal number of training samples (equation 6), the authors introduce the annotation cost $\alpha$ (line 274). No mention of what this value should be is provided, yet having $\alpha < a \cdot b$ implies that a negative training set size is ideal.

1. The paper would benefit from having more empirical evidence. Experiments are conducted on a single dataset and a limited number of architectures. Confidence intervals across model initializations are not reported, which would help provide insight into the method’s robustness.

1. Further assessment of how well the method interpolates as well as extrapolates to unseen entropy scores for increasing dataset sizes (Figure 4) would strengthen the empirical results.

1. The results of this paper and the developed methodology rest on the assumption that entropy decays as the training set size grows. The paper would benefit from discussing under what settings the assumptions about decaying break down and what implications that might have. As far as I understand, suppose a model $M_{t2}$ trained on $D_{t2}$ where $|D_{t2}| = 200$. Say that $k=10$ bins are chosen and we get IoU scores consistently in the range $[0.5, 0.6)$. Then the corresponding probabilities would become: $p_6 = 1$ and all other probabilities would be zero. The entropy $H(p^{(t2)})$ would thus become 0.

1. The paper lacks one or more simple baselines with which the developed method can be compared. I think their methodology is comparable to the literature on experimental design, where one would want to understand the performance of an unknown system, as e.g described in the book “Design and Analysis of Experiments by Montgomery”.

1. The figures simply have too small font sizes; they are not at all readable on printouts, and you have to zoom heavily to read the text.

1. There is a notable amount of spelling mistakes, which makes reading the manuscript less enjoyable.

**Questions:**

1. In L111-112, "We therefore review studies on classification models and large language models (LLMs)" - why is LLM relevant for your approach when you discuss image segmentation models?

1. In L133-134, "entropy has not been used to evaluate or compare medical image segmentation models", while this is probably correct, entropy has been extensively used in active learning approaches for image segmentation models. You may want to include references to relevant literature.

1. L158: "performance difference is typically insignificant if both models are trained on the same dataset.", You should include a reference for this statement, as I believe that model performance may vary significantly from training run to training run.

1. Equation 6: What are reasonable values for the cost of labeling a data point $\alpha$?

1. Table 2: Is it correctly understood that $c$ is the limit of performance for a given model ($n\rightarrow\infty$)? If so, SegFormer-B1 is the better model compared to B0 and B2, but that contradicts the previous claim that performance always improves with model+data size (assuming that B2 is bigger than B1).

1. Table 3: Can confidence intervals be reported as well?

1. Figure 3: How sensitive are the observed and estimated probabilities and entropies to the number of IoU bins chosen?

1. Figure 4: Can the authors extend the results to include more future points or interpolations? Having a single observation provides little evidence as to whether the fitted functional form can reasonably predict unseen observations.

---

### Note · Authors · 2025-11-17

**Comment:**

We appreciate the reviewers’ helpful comments and suggestions. Since additional dataset results are requested and we are unable to produce them within the current timeframe, we will withdraw the submission at this stage.

**Withdrawal Confirmation:**

I have read and agree with the venue's withdrawal policy on behalf of myself and my co-authors.